# Recognition & management of varicella infections and accuracy of antimicrobial recommendations: Case vignettes study in the US

Jaime Fergie[1], Manjiri Pawaskar[2]*, Phani Veeranki[3], Salome Samant[2], Carolyn Harley[3], Joanna MacEwan[3], Taylor T. Schwartz[4], Shikha Surati[2], James H. Conway[5]

1 Driscoll Children's Hospital, Corpus Christi, Texas, United States of America, 2 Merck & Co. Inc., Rahway, New Jersey, United States of America, 3 PRECISIONheor, Los Angeles, California, United States of America, 4 Avalere Health, Washington DC, District of Columbia, United States of America, 5 School of Medicine and Public Health, University of Wisconsin, Madison, Wisconsin, United States of America

* manjiri.pawaskar@merck.com

## Abstract

### Background

In 1995, the CDC recommended one-dose routine varicella immunization for children <12 years of age, expanding its recommendation to two doses in 2006. Today, with widespread varicella vaccination coverage, an estimated 3.5 million cases of varicella, 9,000 hospitalizations, and 100 deaths are prevented annually in the United States. Since varicella infections are now uncommon, health care providers (HCPs) may not recognize varicella infections and may prescribe inappropriate treatment.

### Methods

An online survey of HCPs was conducted to assess recognition and management of varicella infections. Responses to eight varicella vignettes describing patients with varying varicella symptoms were analyzed and descriptive analyses performed. Stratified analysis comparing responses of those licensed before and in/after 1996 was also performed.

### Results

153 HCPs (50 nurse practitioners, 103 doctors) completed the survey. Mean age of respondents was 44 years. 62% were female, and 82% were licensed before 1996. Varicella infection was correctly diagnosed 79% of the time. HCPs correctly recognized uncomplicated varicella vignettes 85% of the time versus 61% of the time for complicated varicella vignettes. Antibiotics were recommended 17% of the time and antivirals 18% of the time, of which 25% and 69% (respectively) were not appropriate per guidelines. HCPs licensed before 1996 were better able to recognize varicella compared to those licensed later, but prescribed more antimicrobials medications to treat varicella.

**Data Availability Statement:** All relevant data are within the manuscript and its Supporting information files.

**Funding:** Received financial support from Merck Sharp & Dohme Corp., a subsidiary of Merck & Co., Inc., Kenilworth, NJ, USA for the execution of this research. T. Schwartz reports personal fees from Merck as a previous employee of PRECISIONheor, during the conduct of the study and personal fees and other from Life Sciences Companies, outside the submitted work. J. H. Conway reports grants and personal fees from Sanofi Pasteur, Pfizer, Merck, GSK, and Centers for Disease Control outside of the submitted work. J. Fergie reports personal fees from Merck Sharp & Dohme Corp, outside the submitted work.

**Competing interests:** This work was supported by Merck Sharp & Dohme Corp., a subsidiary of Merck & Co., Inc., Kenilworth, NJ, USA. M. Pawaskar, S. Samant, and S. Surati are employees of Merck Sharp & Dohme Corp., a subsidiary of Merck & Co., Inc., Kenilworth, NJ, USA, and stockholders of Merck & Co., Inc., Kenilworth, NJ, USA. This does not alter our adherence to PLOS ONE policies on sharing data and materials.

## Conclusions

Although most HCPs recognized varicella infection, a sizable proportion could not recognize cases with complications, and some of the varicella cases were inappropriately treated with antibiotics and/or antivirals. Additional HCP training and high vaccination coverage are important strategies to avoid inaccurate diagnoses and minimize unnecessary exposure to antimicrobial/antiviral therapies.

## Introduction

Varicella infection was common in the United States (US) in the pre-vaccination era, with 4 million varicella cases, 10,000 hospitalizations, and 100 deaths related to varicella occurring annually in US children [1]. Introduction of the varicella vaccination program in 1995 was a major milestone in the fight against this disease, with the Advisory Committee on Immunization Practices (ACIP) recommending 1 dose for routine childhood varicella vaccination and catch-up for children up to 12 years, and 2 vaccine doses for older susceptible persons at greater risk ($\geq$ 13 years of age) [2]. In 2006, ACIP expanded the recommendation to two doses for all children, and adolescents/adults without evidence of immunity [2]. Fifteen years of vaccination (1996 to 2010) allowed the US to demonstrate a dramatic decrease in the annual burden of disease with 20 varicella related deaths (90% decrease compared to pre-vaccine era), 1,700 hospitalizations (84% decrease) and 350,000 varicella cases (92% decrease) in 2010 [1]. It is estimated that the varicella vaccination program saves $373 million in direct cost and $1.598 billion in societal costs, annually in the US [3].

Varicella is highly transmissible with 90% of susceptible household contacts developing varicella after exposure [4]. Most varicella cases resolve without complications, especially among the immunocompetent. However, moderate and serious complications can occur, including pneumonia, bacterial superinfections involving skin and soft tissues, cerebella ataxia, encephalitis and in rare cases, may lead to death [5,6]. Though varicella can also occur among the vaccinated (called breakthrough varicella), such infection is milder, of shorter duration, and more likely to be maculopapular (versus typical vesicular) when compared to the unvaccinated, especially after two- dose vaccination [4]. Most cases of varicella infections can be managed with only supportive care [7]. Cases with complications may require more aggressive management, including use of prescription antivirals and/or antibiotics, and occasionally hospitalization [8]. The American Academy of Pediatrics (AAP) notes that antivirals are primarily recommended for use with populations at increased risk for complications, including immunocompromised individuals [9].

Despite the high rate of vaccination and resulting steep decline in varicella cases over the past several decades, growing hesitancy to childhood vaccinations [10] and delays in vaccinations due to the current COVID-19 pandemic, raise concerns about a potential increase in vaccine preventable diseases, including varicella, in the coming years [4,10,11]. Yet, because of its rarity today, many health care providers (HCPs) who began working after the advent of widespread vaccination may neither recognize varicella infection nor readily implement appropriate treatment. Given the current dynamics, we undertook a study to evaluate the extent to which today's HCPs recognize and manage varicella infections among children in the US.

## Methods

An online survey of HCPs using a clinical case vignette strategy was conducted. Ten case vignettes were drawn from existing literature [12–14] and developed in consultation with

infectious disease experts. The vignettes included brief case descriptions and had accompanying medical case images. Nine of the 10 vignettes were true varicella infections, while one represented separate skin infection to ensure clinicians were able to accurately diagnosis varicella versus other similar infectious diseases. The vignettes varied in the age (1–15 years), health profiles, clinical presentations, childhood vaccination status, and presence of complications. Correct diagnosis (varicella with or without complications) and appropriate disease management was defined for each vignette using the American Academy of Pediatrics (AAP) Red Book, clinical infectious disease expert input, and literature [14].

HCPs were recruited by a survey vendor specializing in health care provider research with physicians and nurses from their existing panel of healthcare professionals (HCPs) across the United States. Respondents were invited via email to take part in the survey. Inclusion criteria included: either a board-certified physician in Pediatrics/ Family medicine/ Internal Medicine specialties, treating at least 100 pediatric patients per month or a licensed nurse practitioner; currently practicing in the United States with ≥ 50% time spent in a clinical setting; English proficiency; and consent to participate. The survey was administered through an on-line portal. All participants received a modest incentive for their participation and were blinded to the study sponsors. Anonymized respondent data was delivered for analysis to prevent responses being linked directly to individual providers. This study was determined to be exempt from Institutional Review Board (IRB) oversight by Advarra IRB; all respondents reviewed and approved an informed consent document on-line prior to completing the survey.

None of the respondents were told that the study was about varicella vaccination, but rather that the survey was about treatment patterns of pediatric patients with various viral and bacterial infections, to avoid respondent bias. Respondents were presented with each case vignette, including a brief description of the case, the age of the child, childhood vaccination status, and an image of the skin lesions and asked to select a diagnosis from a pre-populated list of potential skin infections including many with similar clinical presentations (Table 1a). If a respondent selected varicella, they were then asked to indicate if they believed the case was accompanied by any complication. Following diagnosis, respondents were presented with a list of potential disease management strategies ranging from supportive care to inpatient hospitalization, though only antiviral and antibiotic treatment is in scope for this manuscript. Respondents were permitted to select more than one management or treatment approach (Table 1b). Additionally, information on the respondents' training, working environment, age, gender, and other demographics were collected. Both physicians and nurse practitioners responded to the same survey. Respondent data were analyzed descriptively overall and stratified by whether the HCPs were licensed prior to or after the introduction of the varicella vaccine in 1995.

The final eight vignettes are shown in Table 2 and the survey provided in the supplement. Two of the 10 vignettes were excluded from the final analysis (and from Table 2; details are provided in the supplement) after consultation with two clinicians/ experts. Of the eight varicella cases for which respondent data was evaluated, two of which were defined as having complications. Among the cases, supportive care alone was appropriate for four vignettes. Antibiotics in addition to supportive care were appropriate for two vignettes, and antivirals for another two vignettes (Table 2).

## Results

### Study sample

The study population consisted of 103 physicians, and 50 nurse practitioners, totaling 153 respondents. Mean age of respondents was 43.9 years with the majority being female (62.1%), White (73.9%) and licensed in the post-vaccination era of 1996 or later (81.7%) (Table 3).

**Table 1.** a. List of Potential Diagnoses Options. b. List of Potential Management Strategies *.

| |
|---|
| *Based on the case description above, what do you feel is the most likely primary diagnosis? [Select one]* |
| **Varicella** [*SHOW sub-options if Varicella is selected*] |
| **Varicella without complication** |
| **Varicella with complications** |
| • **Varicella meningitis** |
| • **Varicella encephalitis and encephalomyelitis** |
| • **Varicella myelitis** |
| • **Varicella pneumonia** |
| • **Varicella keratitis** |
| • **Other varicella complications** |
| **Hand foot and mouth disease** |
| **Bacterial infection NOS** |
| **Pruritus, unspecified** |
| **Impetigo** |
| **Scabies** |
| **Poison oak/ivy** |
| **Molluscum contagiosum** |
| **Folliculitis** |
| **Conjunctivitis** |
| **Cellulitis** |
| **Other** |
| *Based on the case description above, what treatment strategy would you recommend and/or prescribe? [Select all that apply]* ** |
| **Supportive care** |
| **Administer varicella vaccine** |
| **Administer pneumococcal vaccine** |
| **Administer varicella immunoglobulin** |
| **Other catch-up vaccines** |
| **Treatment with antivirals** |
| **Treatment with antibiotics** |
| **Hospitalization** |
| **Manage as outpatient** |
| **Admit for inpatient treatment** |
| **Laboratory tests** |
| **Imaging** |
| **Other** |

*Note: Only the antibiotic and antiviral usage was in scope for this manuscript;

** Please see the supplement for a copy of the survey.

*** Supportive care (select all that apply): Acetaminophen, Ibuprofen, Calamine lotion, Colloidal oatmeal bath, Zinc oxide, Topical antihistamines, Topical corticosteroids, Topical antifungal, Antiseptic cleansers (hydrogen peroxide, chlorhexidine etc.), Topical antibiotics, Warm compresses to relieve itching and pain, Other supportive care.

## Recognition of varicella

When presented with the eight vignettes, 79.2% of respondents accurately recognized varicella infection. (Table 4) Those licensed in the pre-vaccination era were more likely to recognize varicella infection (88.8%) compared to those licensed in the post-vaccination era (77.1%). Among respondents who accurately recognized varicella infection, the majority (86.7%)

**Table 2. Summary of case vignettes**[*].

| Vignette | Age and health status | Varicella vaccination status | Diagnosis | Primary Treatment Recommendation [***] |
|---|---|---|---|---|
| 1 | Healthy 3.5-year-old | **Unvaccinated**, no known exposure to varicella | Varicella with no complications | Supportive care |
| 2 | Healthy 5-year-old | **1-dose vaccination**, exposed 14 days prior | Varicella with no complications | Supportive care |
| 3 | Healthy 7-year-old | **Unvaccinated**, not exposed to varicella | Varicella with **Complication** (*S. aureus* infection of lesions) | **Antibiotics** +/- Supportive care |
| 4 | 3-year-old with history of asthma and on prednisone | **Unvaccinated**, exposed to shingles 7 days prior | Varicella with no complications | **Antivirals** +/- Supportive care |
| 5 | Healthy 10-month-old | **Unvaccinated**, exposure to varicella 12 days prior | Varicella with no complications | Supportive care |
| 6 | Healthy 15-year-old | **Unvaccinated** (personal belief exemption), exposure to varicella 10 days prior | Varicella with no complications | Antivirals +/- Supportive care |
| 7 | Healthy 6-year-old, respiratory sign/symptoms. | **1-dose vaccination**, no known exposure to varicella | Varicella with **Complication** (pneumonia) | **Antibiotics** +**Hospitalization** +/- Supportive care |
| 8 | Healthy 14-year-old | **Unvaccinated** (religious belief exemption), no known exposure to varicella. | Varicella with no complications | Supportive care |

[*]Copy of the survey provided in the supplement

[**] 2 vignettes were removed from the analysis after consultation with the clinical consultants due to concerns about unclear treatment option based on vignettes to avoid potential confusion.

[***] The scope of the manuscript was restricted to correct antibiotic and antiviral usage regarding the treatment option.

accurately categorized the infection as uncomplicated or complicated. Respondents licensed in the pre-vaccination era correctly recognized complication status 87.4% of the time, similar to those licensed in the post-vaccination era (86.5%). Among the 254 responses where varicella infection was not correctly identified, bacterial infection, not otherwise specified (24.8%), "other" (20.1%), impetigo (19.3%), and hand, foot & mouth disease (12.2%) were the most common diagnoses selected.

**Table 3. Respondent descriptions.**

| | Licensed prior to 1996 | Licensed in 1996 or later | Total (N = 153) |
|---|---|---|---|
| Provider type | | | |
| Physician | 22 (21.4%) | 81 (78.6%) | 103 (67.3%) |
| Nurse Practitioner | 6 (12.0%) | 44 (88.0%) | 50 (32.7%) |
| Age, mean (SD) | 59.8 (6.8) | 40.4 (7.9) | 43.9 (10.9) |
| N missing | 3 | 13 | 16 |
| Gender | | | |
| Female | 13 (46.4%) | 82 (65.6% | 95 (62.1%) |
| Male | 15 (53.6%) | 41 (32.8%) | 56 (36.6%) |
| I prefer not to answer | 0 (0.0%) | 2 (1.6%) | 2 (1.3%) |
| Race | | | |
| Asian | 2 (7.1%) | 15 (12.0%) | 17 (11.1%) |
| Black/African American | 0 (0.0%) | 6 (4.8%) | 6 (3.9%) |
| White | 25 (89.3%) | 88 (70.4%) | 113 (73.9%) |
| 2+ races | 0 (0.0%) | 2 (1.6%) | 2 (1.3%) |
| Don't know/Not sure | 0 (0.0%) | 1 (0.8%) | 1 (0.7%) |
| Prefer not to answer | 1 (3.6%) | 13 (10.4%) | 14 (9.2%) |

**Table 4. Varicella diagnosis and categorization.**

| | COLUMN A: Vignettes where Varicella was correctly diagnosed* | | | COLUMN B: Vignettes where presence or absence of complications was correctly identified** | | |
|---|---|---|---|---|---|---|
| | All Responses# | Licensed <1996 | Licensed 1996-present | All Responses | Licensed <1996 | Licensed 1996-present |
| | N (%) | N (%) | N (%) | N (%) | N (%) | N (%) |
| **All Responses (8 vignettes)** | **970 (79.2%)** | **199 (88.8%)** | **771 (77.1%)** | **841 (86.7%)** | **174 (87.4%)** | **667 (86.5%)** |
| Varicella vignettes with no complications (6 vignettes) | 784 (85.4%) | 159 (94.6%) | 625 (83.3%) | 167 (89.8%) | 136 (85.5%) | 538 (86.1%) |
| Varicella vignettes with complications (2 vignettes) | 186 (60.8%) | 40 (71.4%) | 146 (58.4%) | 674 (86.0%) | 38 (95.0%) | 129 (88.4%) |
| Vignette 1—Varicella with no complications | 125 (81.7%) | 26 (92.9%) | 99 (79.2%) | 112 (89.6%) | 25 (96.2%) | 87 (87.9%) |
| Vignette 2—Varicella with no complications | 128 (83.7%) | 25 (89.3%) | 103 (82.4%) | 120 (93.8%) | 23 (92.0%) | 97 (94.2%) |
| Vignette 3—Varicella with complication (*Staphylococcus aureus* infection complication of lesion) | 64 (41.8%) | 15 (53.6%) | 49 (39.2%) | 45 (70.3%) | 13 (86.7%) | 32 (65.3%) |
| Vignette 4—Varicella with no complications | 123 (80.4%) | 27 (96.4%) | 96 (76.8%) | 65 (52.8%) | 11 (40.7%) | 54 (56.3%) |
| Vignette 5—Varicella with no complications | 133 (86.9%) | 27 (96.4%) | 106 (84.8%) | 130 (97.7%) | 25 (92.6%) | 105 (99.1%) |
| Vignette 6—Varicella with no complications | 144 (94.1%) | 28 (100%) | 116 (92.8%) | 134 (93.1%) | 28 (100%) | 106 (91.4%) |
| Vignette 7—Varicella with complication (pneumonia) | 122 (79.7%) | 25 (89.3%) | 97 (77.6%) | 122 (100%) | 25 (100%) | 97 (100%) |
| Vignette 8—Varicella with no complications | 131 (85.6%) | 26 (92.9%) | 105 (84.0%) | 113 (86.3%) | 24 (92.3%) | 89 (84.8%) |

* N refers to total number of vignettes and not the number respondents. Thus, the total number of vignettes in the survey is 153 respondents * 8 vignettes each = 1224 vignettes. Of these, only the ones correctly diagnosed are reported here. There was no missing vignette for any respondent.

** Percentage out of those who correctly diagnosed varicella in Column A.

For varicella vignettes with no complications, correct recognition of varicella infection ranged from 80.4% to 94.1%. Once identified as varicella, most respondents were able to categorize the cases as lacking complications (89.6% to 97.7%), except for vignette #4, where nearly half incorrectly believed the case had complications (47.2%). Varicella cases with complications were more difficult to identify as varicella infection. For vignette #3, a varicella with complication (*Staphylococcus aureus* infection), only 41.8% of respondents recognized varicella infection accurately; among those accurately diagnosing varicella, 70.3% identified the vignette as having a varicella related complication. For vignette #7, a varicella with complication (pneumonia), 79.7% of respondents recognized varicella infection accurately; 100% of all respondents who recognized varicella infection accurately identified the case as having complications.

## Treatment patterns: Use of antivirals and antibiotics

The treatment patterns regarding the use of antivirals and antibiotics for the varicella vignettes with and without complications are shown in Table 5.

## Appropriate antiviral and antibiotic use

While 75.4% of antibiotic recommendations were correctly made for two vignettes with complications of varicella, 24.6% of antibiotic recommendations were incorrectly assigned to cases where either supportive care or antiviral therapy were appropriate. Of all antiviral recommendations, only 30.6% were appropriate to the case described (Fig 1).

HCPs licensed in the pre-vaccination era recommended antibiotics and antivirals more often as compared to those licensed in later years (22.8% and 32.1% versus 12.4% and 12.0%). Antibiotics were incorrectly recommended 31.4% of the time by those licensed in the pre-

**Table 5. Use of antivirals and antibiotics for vignettes with and without complications.**

| Antiviral and antibiotic Treatment patterns | Varicella without complications* | | Varicella with complications** | |
|---|---|---|---|---|
| | Actual survey response | Expected use per guidelines | Actual survey response | Expected use per guidelines |
| **Proportion prescribed antivirals** | 15.9% | 50.0% $ | 14.1% | 0.0% |
| **Proportion prescribed antibiotics** | 5.0% | 0.0% | 41.5% | 50.0% $$ |
| **Proportion prescribed both antibiotics and antiviral** | 0.4% | 0.0% | 8.5% | 0.0% |
| **Proportion prescribed neither antibiotics and antiviral** | 78.6% | 50.0% | 35.9% | 50.0% |

*Varicella without complications (From vignettes # 1,2,4,5,6,8);

** Varicella with complications (From Vignettes# 3 and 7).

$ Vignettes 4 and 6 needed antivirals;

$$ Vignettes 3 and 7 needed antibiotics.

vaccination era compared to 22.4% of the time among those licensed in the post-vaccination era (Fig 2). Conversely, antivirals were incorrectly recommended less often by HCPs licensed earlier (63.9%) compared to those licensed later (72.1%).

## Discussion

This study examined diagnostic and antiviral/antibiotic prescription practices for varicella infection in the age of widespread vaccination in the US. Many HCPs today have little experience with varicella infection due to rarity of the disease given widespread use of vaccines. However, vaccine hesitancy [15] and delays in routine pediatric vaccination due to COVID-19 [11] has impacted the vaccination patterns and coverage rates. It is important to accurately diagnose and manage pediatric infectious diseases such as varicella. This study demonstrated that while most varicella cases without complications are correctly identified by HCPs,

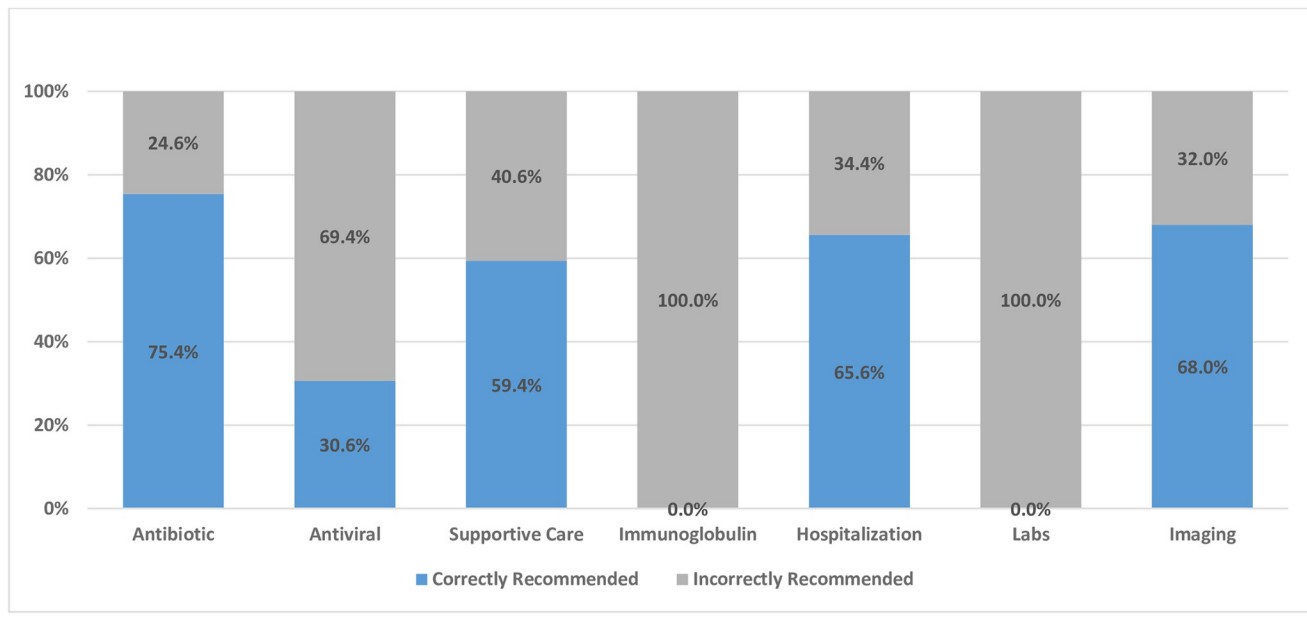

**Fig 1. Percent of correct vs. incorrect treatment recommendations by type of intervention.**

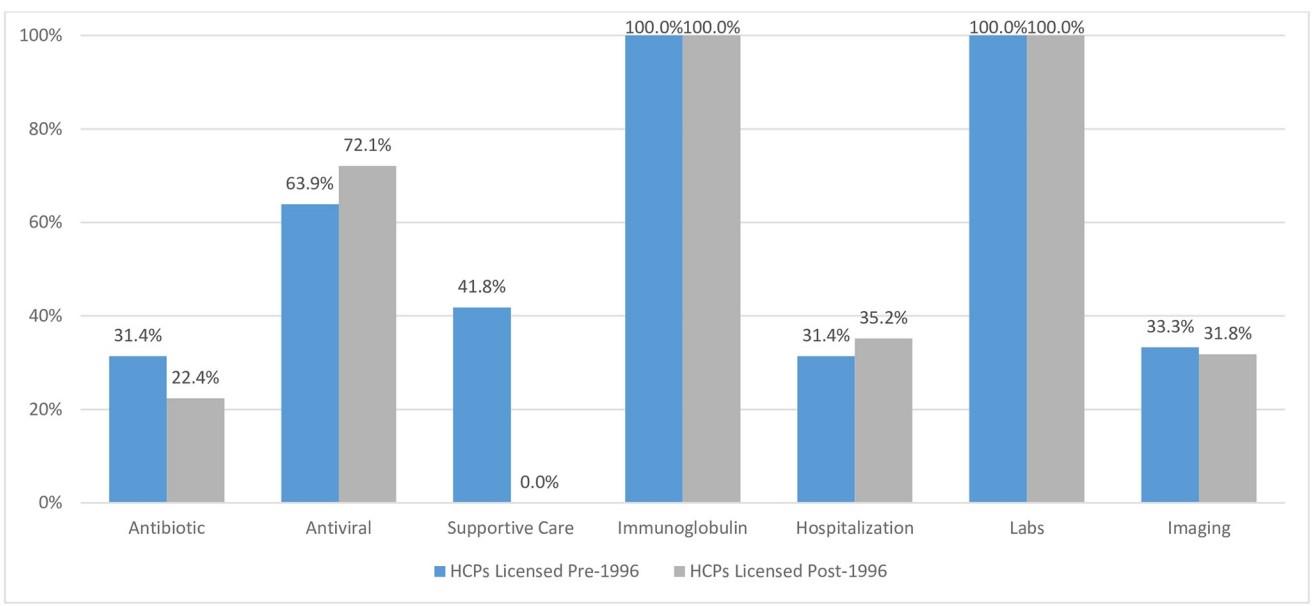

**Fig 2. Percent of incorrect treatment recommendations by type of intervention by year of licensure.**

significantly fewer HCPs were able to correctly recognize varicella when presented with complications.

Our study also showed that some varicella cases may be inappropriately treated with antibiotics and antivirals. The CDC estimates that one in three antibiotic prescriptions in the United States is unnecessary [16] Other countries also show high use of antibiotic use in varicella [8]. While our research does not try to evaluate antibiotic resistance directly, inappropriate use of antibiotics for management of varicella may further contribute to antimicrobial resistance, which is an on-going public health challenge [17,18]. It must be noted that in large part, HCPs accurately assigned management approaches for most vignettes. Supportive care was appropriately recommended as the primary approach for most cases.

Of particular interest is the finding that HCPs trained prior to widespread vaccination were better able to recognize varicella infection in patients and correctly determine presence of complications, but prescribed antibiotics and antivirals more often than recommended. This finding is consistent with previous studies investigating antibiotic prescribing patterns which have demonstrated that older HCPs prescribed a higher volume of antibiotics [19,20].

While this study points to important diagnostic and antimicrobial treatment patterns for varicella infection in an era of widespread vaccination, the study does face certain methodological limitations. Due to limited number of varicella infections in the US, we could not assess current treatment patterns for varicella infection from secondary databases. Hence, we utilized a case vignette approach for this study (8 vignettes). We did not use qualitative methodology for vignette selection and as such the vignettes may not comprehensively represent all varicella cases in the US. However, all vignettes were developed based upon expert clinical consultation and literature review of varicella cases. Unlike standard patient-provider interactions, in vignette studies, patients are unable to communicate their symptoms and providers are unable to ask probing questions. The smaller sample size does not imply generalizability to broader population. Like any other vignette survey, this study is subject to responder bias. Since the majority of respondents were licensed in more recent years, our ability to compare differences more robustly in diagnosis and treatment approaches by HCPs licensed in the pre- and post-

vaccination era was limited. While year of licensure may be considered to be a reasonable proxy for provider experience with varicella, especially considering the sharp decline in varicella cases post vaccination, it is possible that providers licensed after 1996 may have gained experience with varicella cases in mini outbreaks in the country or through global experience. Lastly, our analyses are descriptive only, and the statistical significance of differences between the groups could not be assessed.

Various strategies can help improve the correct diagnosis and recognition of varicella infection as well as prevent inappropriate use of antibiotics and antivirals. Continuing education programs for HCPs on the recognition of varicella, its complications, high-risk groups for more severe disease, and appropriate management of cases (aimed at improving appropriate use of antibiotics and antivirals) could potentially lower healthcare resource utilization and the economic burden of varicella. Ensuring high vaccination coverage despite challenges by COVID-19 is critical.

## Conclusions

In spite of the low incidence of varicella infections in the US, majority of the clinicians accurately diagnosed varicella infection. However, a considerable proportion of clinicians incorrectly recognized varicella infection with complications as well as recommended the inappropriate use of antimicrobial agents for management of varicella. Additional training may help HCPs better recognize and manage cases of varicella infection, while ensuring high rates of varicella vaccination is an important strategy to minimize unnecessary prescribing of antibiotic or antiviral therapies.

## Supporting information

**S1 File. Survey.** Vignette information and survey file.
(DOCX)

## Author Contributions

**Conceptualization:** Jaime Fergie, Manjiri Pawaskar, Phani Veeranki, Salome Samant, Joanna MacEwan, Taylor T. Schwartz, Shikha Surati, James H. Conway.

**Data curation:** Phani Veeranki, Joanna MacEwan, Taylor T. Schwartz.

**Formal analysis:** Carolyn Harley.

**Methodology:** Manjiri Pawaskar, Phani Veeranki, Salome Samant, Joanna MacEwan, Taylor T. Schwartz, Shikha Surati.

**Project administration:** Carolyn Harley.

**Supervision:** Manjiri Pawaskar, Salome Samant, Shikha Surati.

**Validation:** Jaime Fergie, James H. Conway.

**Writing – original draft:** Carolyn Harley, James H. Conway.

**Writing – review & editing:** Jaime Fergie, Manjiri Pawaskar, Phani Veeranki, Salome Samant, Carolyn Harley, Joanna MacEwan, Taylor T. Schwartz, Shikha Surati.

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
