## [Decision Letter · Decision Letter 0]

19 Oct 2021

PONE-D-21-13847Recognition & Management of Varicella Infections and Accuracy of Intervention Recommendations: a case vignettes study in the USPLOS ONE

Dear Dr. Harley,

Thank you for submitting your manuscript to PLOS ONE. We apologize for the delay in our response due to the need for an editor reassignment. After careful consideration, we feel that it has merit but does not fully meet PLOS ONE’s publication criteria as it currently stands. Therefore, we invite you to submit a revised version of the manuscript that addresses the points raised during the review process.

Three reviewers reviewed and provided feedback on your manuscript. The research was felt to be interesting and useful. However, all reviewers agree that the work required extensive reviews before it could be accepted for publication. Most notably, reviewers felt that the data underlying the findings in the manuscript were not made fully available to the reader. Specifically, the methods require much more detail regarding the survey and how the survey was performed. Survey materials should be provided as supplemental material.  On the other hand, the reviewers felt that the introduction, tables, and figures provided less useful information and could be streamlined. References require detailed review. 

We look forward to receiving your revised manuscript.

Kind regards,

Anne Lachiewicz, MD, MPH

Academic Editor

PLOS ONE

Journal Requirements:

2. Thank you for stating in the text of your manuscript "This study was determined to be exempt from Institutional Review Board oversight by Advarra

IRB; all respondents completed an informed consent prior to participating in the study." Please ensure that you have specified what type of consent you obtained (for instance, written or verbal, and if verbal, how it was documented and witnessed). Please also add all of this this information to your ethics statement in the online submission form.

3. Thank you for stating the following financial disclosure: "This work was supported by Merck Sharp & Dohme Corp., a subsidiary of Merck & Co., Inc., Kenilworth, NJ, USA.  "

We note that one or more of the authors is affiliated with the funding organization, indicating the funder may have had some role in the design, data collection, analysis or preparation of your manuscript for publication; in other words, the funder played an indirect role through the participation of the co-authors. If the funding organization did not play a role in the study design, data collection and analysis, decision to publish, or preparation of the manuscript and only provided financial support in the form of authors' salaries and/or research materials, please do the following:

a. Review your statements relating to the author contributions, and ensure you have specifically and accurately indicated the role(s) that these authors had in your study. These amendments should be made in the online form.

b. Confirm in your cover letter that you agree with the following statement, and we will change the online submission form on your behalf: 

“The funder provided support in the form of salaries for authors [insert relevant initials], but did not have any additional role in the study design, data collection and analysis, decision to publish, or preparation of the manuscript. The specific roles of these authors are articulated in the ‘author contributions’ section.

"M. Pawaskar, S. Samant, and S. Surati are employees of Merck Sharp & Dohme Corp., a subsidiary of Merck & Co., Inc., Kenilworth, NJ, USA, and stockholders of Merck & Co., Inc., Kenilworth, NJ, USA. P. Veerkani, C. Harley, and J. MacEwan are employees of PRECISIONheor, which received financial support from Merck Sharp & Dohme Corp., a subsidiary of Merck & Co., Inc., Kenilworth, NJ, USA for the execution of this research. 

T. Schwartz reports personal fees from Merck as a previous employee of PRECISIONheor, during the conduct of the study and personal fees and other from Life Sciences Companies, outside the submitted work. J. H. Conway reports grants and personal fees from Sanofi Pasteur, Pfizer, Merck, GSK, and Centers for Disease Control outside of the submitted work.  J. Fergie reports personal fees from Merck Sharp & Dohme Corp, outside the submitted work."

We note that you received funding from a commercial source: Merck & Co.,PRECISIONheor, Sanofi Pasteur, Pfizer, Merck, GSK

Reviewers' comments:

Reviewer's Responses to Questions

**Comments to the Author**

1. Is the manuscript technically sound, and do the data support the conclusions?

Reviewer #1: Partly

Reviewer #2: Yes

Reviewer #3: Yes

2. Has the statistical analysis been performed appropriately and rigorously? 

Reviewer #1: Yes

Reviewer #2: N/A

Reviewer #3: Yes

3. Have the authors made all data underlying the findings in their manuscript fully available?

Reviewer #1: No

Reviewer #2: No

Reviewer #3: No

4. Is the manuscript presented in an intelligible fashion and written in standard English?

Reviewer #1: Yes

Reviewer #2: Yes

Reviewer #3: Yes

5. Review Comments to the Author

**Reviewer #1:** In this paper, the authors describe a survey based study to evaluate the ability of pediatric physicians and advanced practice providers to recognize varicella in case vignettes. The hypothesis is that vaccination has made cases relatively rare and so cases may be missed when they present for medical attention. While I agree with the premise of the paper, I think there is much more information that needs to be provided before this paper should be published.

First, there is insufficient detail provided about how the survey was done. How were potential participants contacted (what listserv was used?), was this a regional or national group? was an incentive provided?

Second, the survey materials need to be provided as supplemental material for review prior to publication and also for review by any readers who want to access them.

Third, while age might be a very rough correlate, there was no question on whether a provider had previously diagnosed varicella before in their career. There are young people who have worked globally or helped care for those in mini-outbreaks in the US who have good experience with varicella patients. Since this cannot be done, it should be included as a limitation in the discussion.

The introduction does not have to provide the entire background of varicella. It could be streamlined significantly to offer enough detail to justify the study.

I am not sure the figures really offer any additional value to the reader.

**Reviewer #2: **Overall, I think it was a nicely done study looking at an important questions. I found the methods a little weak in how the vignettes, their definitions, and management plans were chosen. The results have so many numbers that it is difficult to pull out the important findings while reading. I think much of this results from whether the denominator is the providers, the vignettes, or the treatment options – as this changes throughout the paper -- it is difficult to go back and forth between these.

Intro –page 3, it would be useful to have some case numbers so that the reader could appreciate the absolute reductions along with the percentage drops – at least for conditions with appreciable morbidity (ie hospitalizations, pneumonia, encephalitis)

Is there a more recent case estimate than 2014?

It would be useful to know if any specific qualitative methodology helped to guideline the number/section of cases. Why 9:1 – that seem overly obvious.

Page 6. I would state that the removed varicella case was excluded from table 1.

Did a lot of people struggle with vignette #7? I would think that the correct answer would be hospitalization and antivirals +/- antibiotics for a child with a severe complication of VZV and underlying lung disease.

Table 3. I think should it be NP or APP rather than nurse?

Page 12. What reference are you using to define complicated vs uncomplicated VZV? #4 is really challenging as in many infectious disease states (ie, complicated UTI or complicated Staph aureus bacteremia) the “complicated” can be defined by EITHER the disease state OR the host being complicated. It is easy in #4 to see how the respondents may have considered a patient on an undefined amount of steroids to be an immunocompromised host and therefore “complicated” even if the presentation itself was not complicated. The fact that >50% called this case complicated suggest that the definitions chosen by the authors may not reflect what is considered complicated in clinical practice.

The following phrase is unclear “Among those who did not recognize varicella infection correctly across all vignettes (n=254),” as there were only 153 respondents. I think this means among those vignettes answered incorrectly? 153*8=1224; 254/1224=0.2 (20% or 254 incidents where varicella was incorrectly identified)

Although the table 4 says only 970 case (79.2%) of cases were correctly identified as varicella.

Lots of numbers are reported but there are really too many to take in.

The most interesting finding is that 20% of varicella cases were missed and 40% of complicated cases were misdiagnosed (if I interpreted this correctly).

This sentence is not clear “Across all uncomplicated vignettes, respondents licensed in the pre-vaccination era were more likely to recognize the case as varicella (94.6%) compared to those licensed in later years (83.3%).” Maybe move the ALL? “Across –all- uncomplicated vignettes, respondents licensed in the pre-vaccination era were more likely to recognize ALL cases as varicella (94.6%) compared to those licensed in later years (83.3%).

So <1996, 5% of uncomplicated cases and 29% of uncomplicated cases were misdiagnosed.

>1996, 23% of uncomplicated cases and 41% of complicated cases were misdiagnosed.

The figures are missing critical information. How the percentages were calculated should be self-evidence when one looks at a figure rather than having to go back to the text to figure out what the authors were using to calculate the percentages. For example, looking at the figure 1b, it is difficult to know whether labs were offered every time and selected incorrectly 100% percent of those times or whether every respondent selected it incorrectly at least once.

Again, I think presenting the #/% missed diagnoses rather than the correct diagnoses would have a great impact on the reader.

**Reviewer #3: **This is an original, interesting, and useful study. I encourage the authors to pursue its publication because it has practical implications for developing communication messages for a disease that is vaccine-preventable. Having said that, I identified several issues during my review. As general comments, there are many inaccuracies in the introduction that need to be fixed, I had some concerns about the primary treatment recommendations for some of the vignettes that influence the results, some wording in the results that seem to indicate statistical significance although the authors report that they did not assess it. In general, I think it would be beneficial to work with an editor for the next revision.

Specific comments are below:

Intro

- 1st para, 2nd sentence: single dose was the routine recommendation for children but people age 13+ years were recommended 2 doses. What is indicated as “single dose vaccination for individuals 12 months of age and older” is not accurate as indicated above.

- 1st para, 4th sentence: I don’t have the 2015 edition of the Pink Book as the 2021 edition is now available online but what the authors indicate as % reduction in this sentence seems to be vaccine coverage. I am not aware of reports of incidence trends reported for these specific age-groups, 19-35 months and 13-17 years, these are the ages where vaccination status is routinely reported. Please check and revise.

- Ref 4: it’s a paper on neurologic complications of herpes zoster, I could not find anything about the 2-dose regimen for varicella vaccination. The appropriate ref is the 2007 MMWR with ACIP recommendations. In the same sentence the vaccine “suppresses cases of varicella” is unusual, better “prevents”. Ref 4 is also inaccurately cited at the end of the 3rd para for varicella complications which the article does not present.

- Ref 5: described trends in outbreaks and not in cases as the authors indicated. The appropriate ref here is Lopez MMWR 2016 (current ref 9).

- 2nd para, 1st sentence: coverage is reported above, no need to mention it here again. “annual incidence remains as 3.9/100,00 ….”, remains is misleading, this is really a low rate and as reported in ref 9 it was an 85% decline from 2006, with continuing decline over that interval. Ref 8: cannot tell what ref is this -- Prevention CfDCa. Chickenpox (Varicella): Monitoring the Impact of Varicella Vaccination 2018 [. The same for ref 10 although this ref has a link.

- 3rd para, 3rd sentence: CDC does not make recommendations for treatment. CDC website cites the American Academy of Pediatrics (AAP) recommendations. There is a link that goes to the recommendations for treatment where it is mentioned that AAP recommended. Please update the reference to the original source.

- Last para, 1st sentence: decline in varicella outbreaks was not mentioned before, the intro indicated reduction in cases, should either add “cases” or mention only cases. I think cases are more relevant than outbreaks. “several decades” it’s really 2.5 decades.

- Ref 18: I was not able to find this reference online. The closest I came was a discussion with 2 experts about covid vaccines for children. Can the authors provide the link? Also, there are several articles published that show a show decline in vaccination rates among children during covid, suggest adding a peer-reviewed publication here. Same for the citation in the results.

Methods

- First paragraph: “The vignettes had a variety of uncomplicated and complicated presentations “, the vignettes do not have complications, maybe they presented or described, please edit. Also use of complicated and uncomplicated as adjectives vs varicella presentations with complications and without complications.

- Vignettes – Recommendations: unclear what “+/- catch-up vaccines” means? Is it that it can be recommended or not, and assessment should be based on what? Is this varicella vaccine or other vaccines? An acute disease is a precaution for vaccination. Vaccination is not indicated on vignettes 6 and 8, how were these patients different from the others for which vaccination is appropriate management?

- Vignette 7: what is “lung problems“? I would think that antivirals are appropriate for this child given varicella with complications (pneumonia), pulmonary problems, and hospitalization.

- Vignette 8: why antivirals are not considered appropriate for this case? It’s a 14-year-old and per the Red Book -- Oral acyclovir or valacyclovir should be considered for otherwise healthy people at increased risk of moderate to severe varicella, such as unvaccinated people older than 12 years, ….. How are patients on vignette 8 (antivirals not recommended) and 6 (antivirals recommended) different? Both are older than 12 and have varicella; exposure should not matter in this case as they are both older than 12. Red Book does indicate that “some experts recommend use of oral acyclovir or valacyclovir for secondary household cases …”, exposure does not indicate if it was household but even though the 14-year-old would be recommended for antivirals based on age.

Results

- First paragraph: “of these female respondents, 65.6% were licensed in the post-vaccination era.” Is this detail important, info is also in the table. Recommend deleting from text.

- Table 3. Row on Year of licensure is not needed it the table. It is reported in the text and having it in the table does not add anything for the reader.

- “more likely”, “less likely” mean statistical significance. They are used several times in the results, please provide p values or reword.

- Page 12, 1st para: use of “complicated cases” and “uncomplicated cases” makes the reading difficult here. Usually a “complicated case” is a difficult case and the expression should not be used as a substitute for “a case with complications”, at least not in the written literature. The authors used that wording throughout the article. “a complicated varicella infection with Staphylococcus aureus infection” vs. varicella complicated with Staph aureus infection.

- Page 12, 2nd para: “Across all uncomplicated vignettes”, the vignettes cannot be complicated please reword.

- Page 13, 2nd para: Lab tests - unclear what lab tests are these. I think lab test to confirm varicella is appropriate; also, for cases with complications lab tests would be appropriate. The authors indicate that “all were unnecessary”.

Discussion

- 2nd para: maybe para on antibiotic resistance can be moved later, the contribution of varicella would likely be small in the US because varicella is rare now, most cases occur in vaccinated patients and are mild, ¾ of recommendations were correctly made.

- Page 15, 2nd para: message of the last sentence is unclear and possibly inaccurate - If only a small fraction of all cases of varicella lead to hospitalization (e.g., 1% of cases leading to 5000 hospitalizations), this could lead to substantial costs to the health care system ranging from $20.7m to $110.6m. 1) The number of hospitalizations is high, in the prevaccine era an average of 10,500 hospitalizations occurred (Galil, PIDJ 2002) and they declined 93% by 2012 (Leung JPIDS 2015); 2) It seems that the sentence is in the context of hospitalizations due to incorrect recommendations for hospitalization, do the authors imply that 5,000 unnecessary hospitalizations would occur? 3) Unclear what the timeline of the $ estimate is – annual or over how many years. Not sure the authors have enough data to estimate unnecessary hospitalizations based on their study.

- Page 15, last line: seems a little offensive to indicate that clinicians licensed before 1995 are more seasoned clinicians compared to HCP’s licensed after 1995, do the authors imply that they are better clinicians?

6. PLOS authors have the option to publish the peer review history of their article (what does this mean?). If published, this will include your full peer review and any attached files.

Reviewer #1: No

Reviewer #2: No

Reviewer #3: No

---

## [Author Response · Author response to Decision Letter 0]

3 Mar 2022

Review Comments to the Author & Responses

Thank you to all 3 reviewers for taking time to review the paper thoroughly and providing such insightful comments. Please see our responses below. We hope that we have addressed them to your satisfaction.

Best regards,

Authors

Reviewer #1:

In this paper, the authors describe a survey-based study to evaluate the ability of pediatric physicians and advanced practice providers to recognize varicella in case vignettes. The hypothesis is that vaccination has made cases relatively rare and so cases may be missed when they present for medical attention. While I agree with the premise of the paper, I think there is much more information that needs to be provided before this paper should be published.

1) First, there is insufficient detail provided about how the survey was done. How were potential participants contacted (what listserv was used?), was this a regional or national group? was an incentive provided?

Response: Thank you for your comment. We have added details in the Methods section regarding the inclusion criteria, recruitment, and honoraria. The text now reads as follows, “HCPs were recruited by a survey vendor specializing in health care provider research with physicians and nurses from their existing panel of healthcare professionals (HCPs) across the United States. Respondents were invited via email to take part in the survey. Inclusion criteria included: either a board-certified physician in pediatrics/ family medicine/ Internal Medicine specialties treating at least 100 pediatric patients per month or a licensed nurse practitioner; currently practicing in the United States with ≥ 50% time spent in a clinical setting; English proficiency; and consent to participate. The survey was administered through an on-line portal. All participants received a modest incentive for their participation and were blinded to the study sponsors. Anonymized respondent data was delivered for analysis to prevent responses being linked directly to individual providers.”

2) Second, the survey materials need to be provided as supplemental material for review prior to publication and also for review by any readers who want to access them.

Response: We have included a copy of the survey in the supplement.

3) Third, while age might be a very rough correlate, there was no question on whether a provider had previously diagnosed varicella before in their career. There are young people who have worked globally or helped care for those in mini-outbreaks in the US who have good experience with varicella patients. Since this cannot be done, it should be included as a limitation in the discussion.

Response: Thank you for your valuable feedback. We wanted to prevent introduction of any bias by specifically asking about varicella and use of post-1996 licensure was an approximate proxy to reflect that physicians would see fewer varicella cases with each year post introduction of varicella vaccination. This has been now addressed in the discussion section as a limitation as follows:

 “While year of licensure may be considered to be a reasonable proxy for provider experience with varicella, especially considering the sharp decline in varicella cases post vaccination, it is possible that providers licensed after 1996 may have gained experience with varicella cases in mini outbreaks in the country or through global experience. Lastly, our analyses are descriptive only, and the statistical significance of differences between the groups could not be assessed.”

4) The introduction does not have to provide the entire background of varicella. It could be streamlined significantly to offer enough detail to justify the study.

Response: Thank you. We have streamlined content throughout the background section of the manuscript as advised.

5) I am not sure the figures really offer any additional value to the reader.

Response: Thank you. We have removed the 2 figures. In addition, in response to other comments received, we have focused the results on the treatment options to the incorrect usage of antibiotic and antiviral and provided the data in the text.

Reviewer #2

 Overall, I think it was a nicely done study looking at an important question. I found the methods a little weak in how the vignettes, their definitions, and management plans were chosen. The results have so many numbers that it is difficult to pull out the important findings while reading. I think much of this results from whether the denominator is the providers, the vignettes, or the treatment options – as this changes throughout the paper -- it is difficult to go back and forth between these.

Response: Thank you for your valuable feedback. We have streamlined the results, with focus on varicella diagnosis and treatment. Data on the correct diagnosis in the text; with more details for provided in the table 4. Also, as regards the treatment options, we have restricted the scope of the paper to the antibiotic and antiviral usage. Additionally, we have added more details on the vignette’s selection process in the methods and clarification of the numbers/denominators as footnote s to Table 1 and 4. 

1) Intro –page 3, it would be useful to have some case numbers so that the reader could appreciate the absolute reductions along with the percentage drops – at least for conditions with appreciable morbidity (i.e. hospitalizations, pneumonia, encephalitis)

Response: Thank you. We have added the absolute reductions as follows to better capture the impact. “Fifteen years of vaccination (1996 to 2010) allowed the US to demonstrate a dramatic decrease in burden of disease with only 20 annual deaths (90% decrease compared to pre-vaccine era), 1,700 hospitalizations (84% decrease) related to varicella reported in 2010 (1). It is estimated that the varicella vaccination program saves $373 million in direct cost and $1.598 billion in societal costs, annually in the US. (3)” 

We restricted the reported impact to number of cases, hospitalizations, and mortality, but did not go into details for specific complications due to scarcity of data and our efforts to streamline the introduction in response to another reviewer comment.

2) Is there a more recent case estimate than 2014?

Response: This statement has been deleted to tighten the background section.

3) It would be useful to know if any specific qualitative methodology helped to guideline the number/section of cases. Why 9:1 – that seem overly obvious.

Response: We did not use qualitative methodology for vignette selection and this has been added as a part of limitation in the discussion section as follows, “We did not use qualitative methodology for vignette selection and as such the vignettes may not comprehensively represent all varicella cases in the US. However, all vignettes were developed based upon expert clinical consultation and literature review of varicella cases.” However, the primary objective of this study was to assess the ability to diagnosis varicella cases. We didn’t want to introduce respondent bias by informing them about varicella management upfront by disclosing this information.

Additional information has been added to the supplemental information as follows: “The first was a non-varicella case/vignette was included to gauge providers ability to distinguish between conditions with symptoms/presentations similar to varicella that need antibiotics. It had been deliberately added to avoid having all varicella cases. The other was a varicella case but removed after expert consultation since the choice of correct treatment was unclear due to the complexity of the case. Further clinical information, not provided in the vignettes, would have been necessary to make an appropriate treatment recommendation.” 

4) Page 6. I would state that the removed varicella case was excluded from table 1.

Response: Thank you. We have edited the text to read “The final eight vignettes are shown in Table 1 and the survey provided in the supplement. Two of the 10 vignettes were excluded from the final analysis (and from Table 1 ; details are provided in the supplement) after consultation with 2 clinicians/ experts “ and added a footnote to the table. In addition, the supplement has more details on why the 2 cases were excluded. “ 

5) Did a lot of people struggle with vignette #7? I would think that the correct answer would be hospitalization and antivirals +/- antibiotics for a child with a severe complication of VZV and underlying lung disease.

Response: We have corrected a typo here. Apologies.

7 Healthy 6-year-old, respiratory sign/symptoms seen. 1-dose vaccination, no known exposure to varicella Varicella with

Complication (pneumonia) Antibiotics

Hospitalization

+/- Supportive care

We have added an explanation in the supplement as follows: “Similarly, for Vignettes#7 again, the 5-day history of rash in a previously healthy 6 year old child meant that the pulmonary complications were more likely secondary bacterial pneumonia rather than primary viral process(AAP reference). It would be most appropriate to hospitalize and start antibiotics. While some physicians might initiate antivirals for some children while evaluating for possible immunodeficiency, consensus for the appropriate treatment of most children would be hospitalization + antibiotics”

6) Table 3. I think should it be NP or APP rather than nurse?

Response: Thank you. We have corrected this information. The provider type in the table now reads “Nurse Practitioner.”

7) Page 12. What reference are you using to define complicated vs uncomplicated VZV? #4 is really challenging as in many infectious disease states (i.e., complicated UTI or complicated Staph aureus bacteremia) the “complicated” can be defined by EITHER the disease state OR the host being complicated. It is easy in #4 to see how the respondents may have considered a patient on an undefined amount of steroids to be an immunocompromised host and therefore “complicated” even if the presentation itself was not complicated. The fact that >50% called this case complicated suggest that the definitions chosen by the authors may not reflect what is considered complicated in clinical practice.

Response: We have updated the manuscript, so that “uncomplicated” and “complicated” varicella is correctly referred to as “varicella with and without complications”. In addition, we have added a statement in the Methods section stating that “Correct diagnosis (varicella with or without complications) and appropriate disease management was defined for each vignette using the American Academy of Pediatrics (AAP) Red Book, clinical infectious disease expert input, and literature.”

8) The following phrase is unclear “Among those who did not recognize varicella infection correctly across all vignettes (n=254),” as there were only 153 respondents. I think this means among those vignettes answered incorrectly? 153*8=1224; 254/1224=0.2 (20% or 254 incidents where varicella was incorrectly identified) Although the table 4 says only 970 case (79.2%) of cases were correctly identified as varicella.

Response: - Restated for clarity as follows: “Among the 254 responses where varicella infection was not correctly identified, bacterial infection, not otherwise specified (24.8%), “other” (20.1%), impetigo (19.3%), and hand, foot & mouth disease (12.2%) were the most common diagnoses selected.” Also, a footnote was added to table 4 “* N refers to total number of vignettes and not the number respondents. Thus, the total number of vignettes in the survey is 153 respondents*8 vignettes each =1224 vignettes. Of these, only the ones correctly diagnosed are reported here. There was no missing vignette for any respondent.”

9) Lots of numbers are reported but there are really too many to take in.

The most interesting finding is that 20% of varicella cases were missed and 40% of complicated cases were misdiagnosed (if I interpreted this correctly).

Response: We have streamlined the results, choosing to report the antibiotic and antiviral usage for treatment options rather than trying to report everything. Thus, we have deleted the figures

10) This sentence is not clear “Across all uncomplicated vignettes, respondents licensed in the pre-vaccination era were more likely to recognize the case as varicella (94.6%) compared to those licensed in later years (83.3%).” Maybe move the ALL? “Across –all- uncomplicated vignettes, respondents licensed in the pre-vaccination era were more likely to recognize ALL cases as varicella (94.6%) compared to those licensed in later years (83.3%).

Response: To aid in clarity, we have removed the term “all” from the sentence. 

11) So <1996, 5% of uncomplicated cases and 29% of uncomplicated cases were misdiagnosed.>1996, 23% of uncomplicated cases and 41% of complicated cases were misdiagnosed.

Response: Thank you for the suggestion. We have incorporated the suggested language as “Those licensed in the pre-vaccination era were more likely to recognize varicella infection (88.8%) compared to those licensed in the post-vaccination era (77.1%). Among respondents who accurately recognized varicella infection, the majority (86.7%) accurately categorized the infection as uncomplicated or complicated. Respondents licensed in the pre-vaccination era correctly recognized complication status 87.4% of the time, similar to those licensed in the post-vaccination era (86.5%).” 

12) The figures are missing critical information. How the percentages were calculated should be self-evidence when one looks at a figure rather than having to go back to the text to figure out what the authors were using to calculate the percentages. For example, looking at the figure 1b, it is difficult to know whether labs were offered every time and selected incorrectly 100% percent of those times or whether every respondent selected it incorrectly at least once.

Response: Thank you. We have deleted the 2 figures. In addition, in response to other comments received, we have focused the results regarding the treatment options to the incorrect usage of antibiotic and antiviral and provided the data in the text

13) Again, I think presenting the #/% missed diagnoses rather than the correct diagnoses would have a great impact on the reader.

Response: Thank you. We have reported the %missed diagnosis in the text as suggested in the comment above. We have kept the % correct diagnosis in the table 4 since the number of vignettes correctly diagnosed as varicella form the denominator for the % that is diagnosed correctly as complicated or not. 

Reviewer #3

This is an original, interesting, and useful study. I encourage the authors to pursue its publication because it has practical implications for developing communication messages for a disease that is vaccine-preventable. Having said that, I identified several issues during my review. As general comments, there are many inaccuracies in the introduction that need to be fixed, I had some concerns about the primary treatment recommendations for some of the vignettes that influence the results, some wording in the results that seem to indicate statistical significance although the authors report that they did not assess it. In general, I think it would be beneficial to work with an editor for the next revision. Specific comments are below:

RESPONSE: Thank you for your review and taking time for providing such insightful comments. We hope that we have addressed them satisfactorily below.

Intro:

1) 1st para, 2nd sentence: single dose was the routine recommendation for children but people age 13+ years were recommended 2 doses. What is indicated as “single dose vaccination for individuals 12 months of age and older” is not accurate as indicated above.

Response: Thank you. The sentence now has an updated reference for the 2007 ACIP CDC recommendations and reads as follows: “Introduction of the varicella vaccination program in 1995 was a major milestone in the fight against this disease, with the Advisory Committee on Immunization Practices (ACIP) recommending 1 dose for routine childhood varicella vaccination and catch-up for children up to 12 years and 2 vaccine doses for older susceptible persons at greater risk (≥ 13 years of age). (2) In 2006, ACIP expanded to a two-dose recommendation for all children and adolescents/adults without evidence of immunity.”

2) 1st para, 4th sentence: I don’t have the 2015 edition of the Pink Book as the 2021 edition is now available online but what the authors indicate as % reduction in this sentence seems to be vaccine coverage. I am not aware of reports of incidence trends reported for these specific age-groups, 19-35 months and 13-17 years, these are the ages where vaccination status is routinely reported. Please check and revise.

Response: Thank you. We have updated this to the more commonly available reference from the CDC infographic. The text was updated as” Fifteen years of vaccination (1996 to 2010) allowed the US to demonstrate a dramatic decrease in the annual burden of disease with 20 varicella related deaths (90% decrease compared to pre-vaccine era), 1,700 hospitalizations (84% decrease) and 350,000 varicella cases (92% decrease) in 2010 (1). It is estimated that the varicella vaccination program saves $373 million in direct cost and $1.598 billion in societal costs, annually in the US. (3)

3) Ref 4: it’s a paper on neurologic complications of herpes zoster, I could not find anything about the 2-dose regimen for varicella vaccination. The appropriate ref is the 2007 MMWR with ACIP recommendations. In the same sentence the vaccine “suppresses cases of varicella” is unusual, better “prevents”. Ref 4 is also inaccurately cited at the end of the 3rd para for varicella complications which the article does not present

Response: Thank you for noticing this error. The sentence now has an updated reference for the 2007 ACIP CDC recommendations and reads as follows: “Introduction of the varicella vaccination program in 1995 was a major milestone in the fight against this disease, with the Advisory Committee on Immunization Practices (ACIP) recommending 1 dose for routine childhood varicella vaccination and catch-up for children up to 12 years and 2 vaccine doses for older susceptible persons at greater risk (≥ 13 years of age). (2) In 2006, ACIP expanded the recommendation to two doses for all children, and adolescents/adults without evidence of immunity.” 

4) Ref 5: described trends in outbreaks and not in cases as the authors indicated. The appropriate ref here is Lopez MMWR 2016 (current ref 9).

Response: The reference has been updated to the Lopez MMWR 2016 reference.

5) 2nd para, 1st sentence: coverage is reported above, no need to mention it here again. “annual incidence remains as 3.9/100,00 ….”, remains is misleading, this is really a low rate and as reported in ref 9 it was an 85% decline from 2006, with continuing decline over that interval. Ref 8: cannot tell what ref is this -- Prevention CfDCa. Chickenpox (Varicella): Monitoring the Impact of Varicella Vaccination 2018 [. The same for ref 10 although this ref has a link.

Response: We have removed this statement in order to streamline the introduction.

6) 3rd para, 3rd sentence: CDC does not make recommendations for treatment. CDC website cites the American Academy of Pediatrics (AAP) recommendations. There is a link that goes to the recommendations for treatment where it is mentioned that AAP recommended. Please update the reference to the original source.

Response: The sentence now reads as follows “The American Academy of Pediatrics (AAP) notes that antivirals are primarily recommended for use with populations at increased risk for complications, including immunocompromised individuals.” The citation now links to an AAP source directly.

7) Last para, 1st sentence: decline in varicella outbreaks was not mentioned before, the intro indicated reduction in cases, should either add “cases” or mention only cases. I think cases are more relevant than outbreaks. “several decades” it’s really 2.5 decades.

Response: The sentence now reads as follows: “Despite the high rate of vaccination and resulting steep decline in varicella cases over the past several decades, growing hesitancy to childhood vaccinations (10) and delays in vaccinations due to the current COVID-19 pandemic, raise concerns about a potential increase in vaccine preventable diseases, including varicella, in the coming years. (11)”

8) Ref 18: I was not able to find this reference online. The closest I came was a discussion with 2 experts about covid vaccines for children. Can the authors provide the link? Also, there are several articles published that show a show decline in vaccination rates among children during covid, suggest adding a peer-reviewed publication here. Same for the citation in the results.

Response: Thank you. We have added a journal reference for this in the introduction section and deleted the sentence from the Discussion section.

Methods:

9) First paragraph: “The vignettes had a variety of uncomplicated and complicated presentations “, the vignettes do not have complications, maybe they presented or described, please edit. Also use of complicated and uncomplicated as adjectives vs varicella presentations with complications and without complications.

Response: The sentence now reads as follows: “The vignettes varied in the age (1-15 years), health profiles, clinical presentations, childhood vaccination status, and presence of complications.”

10) Vignettes – Recommendations: unclear what “+/- catch-up vaccines” means? Is it that it can be recommended or not, and assessment should be based on what? Is this varicella vaccine or other vaccines? An acute disease is a precaution for vaccination. Vaccination is not indicated on vignettes 6 and 8, how were these patients different from the others for which vaccination is appropriate management?

RESPONSE: Since some of the vignettes did not have up-to date vaccination status, this referred to catching up the patient to ensure he is up to date with all vaccines, when possible. However, analyzing that was not within scope of this analysis.

11) Vignette 7: what is “lung problems“? I would think that antivirals are appropriate for this child given varicella with complications (pneumonia), pulmonary problems, and hospitalization.

Response: We apologize. Vignette 7 in table 1 was corrected to:

7 Healthy 6-year-old, respiratory sign/symptoms. 1-dose vaccination, no known exposure to varicella Varicella with

Complication (pneumonia) Antibiotics

+Hospitalization

+/- Supportive care

 ”We have also added an explanation in the supplement as follows to explain the reasoning behind the treatment pattern: …“AAP Red Book indicates that “Antiviral drugs have a limited window of opportunity to affect the outcome of VZV infection. In immunocompetent hosts, most virus replication has stopped by 72 hours after onset of rash; the duration of replication may be extended in immunocompromised hosts……Similarly, for Vignettes#7 again, the 5-day history of rash in a previously healthy 6 year old child meant that the pulmonary complications were more likely secondary bacterial pneumonia rather than primary viral process(AAP reference). It would be most appropriate to hospitalize and start antibiotics. While some physicians may prefer to start antivirals just in case while working the child up for immunodeficiency, so we feel that the appropriate treatment would be hospitalization + antibiotics”

 A copy of the survey in the supplement has bene provided where the description for this vignette states, “A 6-year-old previously healthy child presented with a 5-day history of pruritic rash, which consisted of approximately 350 vesicular lesions, loss of appetite, fatigue, and chest pain. The patient developed a high-grade fever and cough 4 days after the onset of rash symptoms. On examination, the child’s temperature was 102° F. The child was unable to tolerate oral intake, had a respiratory rate < 50/minute, and decreased oxygen saturation (65%). Auscultation of lungs revealed crackles in lower lobes of lungs. Growth and development were otherwise normal. All of the child’s immunizations were not up to date according to the recommended US child vaccination schedule: the child missed the second doses of the MMR and varicella vaccines. The child had no known exposure to hand foot and mouth disease, scabies, or poison ivy/oak ”

12) Vignette 8: why antivirals are not considered appropriate for this case? It’s a 14-year-old and per the Red Book -- Oral acyclovir or valacyclovir should be considered for otherwise healthy people at increased risk of moderate to severe varicella, such as unvaccinated people older than 12 years, ….. How are patients on vignette 8 (antivirals not recommended) and 6 (antivirals recommended) different? Both are older than 12 and have varicella; exposure should not matter in this case as they are both older than 12. Red Book does indicate that “some experts recommend use of oral acyclovir or valacyclovir for secondary household cases …”, exposure does not indicate if it was household but even though the 14-year-old would be recommended for antivirals based on age.

Response: Thank you. We have now addressed this in the supplement as follows: For vignette#8, after expert consultation, the use of antivirals was deemed unnecessary based on the duration of time (5 days rash) & description of an immunocompetent individual. The 5 days vs 3 days was the critical difference since the AAP Red Book indicates that “Antiviral drugs have a limited window of opportunity to affect the outcome of VZV infection. In immunocompetent hosts, most virus replication has stopped by 72 hours after onset of rash; the duration of replication may be extended in immunocompromised hosts” (AAP reference)

Results

13) First paragraph: “of these female respondents, 65.6% were licensed in the post-vaccination era.” Is this detail important, info is also in the table. Recommend deleting from text.

Response: Agree. We have removed this as suggested.

14) Table 3. Row on Year of licensure is not needed it the table. It is reported in the text and having it in the table does not add anything for the reader.

Response: This row has been removed as suggested.

15) “more likely”, “less likely” mean statistical significance. They are used several times in the results, please provide p values or reword.

Response: Instances where “more likely” and “less likely” were used in the paper have been removed and replaced with more specific language, such as “prescribed more often.” 

16) Page 12, 1st para: use of “complicated cases” and “uncomplicated cases” makes the reading difficult here. Usually a “complicated case” is a difficult case and the expression should not be used as a substitute for “a case with complications”, at least not in the written literature. The authors used that wording throughout the article. “a complicated varicella infection with Staphylococcus aureus infection” vs. varicella complicated with Staph aureus infection.

Response: Thank you. We have revised the manuscript accordingly in both the text and tables. We hope that it meets your satisfaction.

17) Page 12, 2nd para: “Across all uncomplicated vignettes”, the vignettes cannot be complicated please reword.

Response: Thank you. The statement was deleted as part of the streamlining the results. But the manuscript was revised as per he suggested advice regarding correct use of the terms.

18) Page 13, 2nd para: Lab tests - unclear what lab tests are these. I think lab test to confirm varicella is appropriate; also, for cases with complications lab tests would be appropriate. The authors indicate that “all were unnecessary”.

Response: We revised the scope of the reported treatment to antiviral and antibiotics only.

Discussion

19) 2nd para: maybe para on antibiotic resistance can be moved later, the contribution of varicella would likely be small in the US because varicella is rare now, most cases occur in vaccinated patients and are mild, ¾ of recommendations were correctly made.

Response: The second and third paragraphs have been swapped.

20) Page 15, 2nd para: message of the last sentence is unclear and possibly inaccurate - If only a small fraction of all cases of varicella lead to hospitalization (e.g., 1% of cases leading to 5000 hospitalizations), this could lead to substantial costs to the health care system ranging from $20.7m to $110.6m. 1) The number of hospitalizations is high, in the prevaccine era an average of 10,500 hospitalizations occurred (Galil, PIDJ 2002) and they declined 93% by 2012 (Leung JPIDS 2015); 2) It seems that the sentence is in the context of hospitalizations due to incorrect recommendations for hospitalization, do the authors imply that 5,000 unnecessary hospitalizations would occur? 3) Unclear what the timeline of the $ estimate is – annual or over how many years. Not sure the authors have enough data to estimate unnecessary hospitalizations based on their study.

Response: Thank you. We have deleted this statement to minimize confusion and streamline the discussion.

21) Page 15, last line: seems a little offensive to indicate that clinicians licensed before 1995 are more seasoned clinicians compared to HCP’s licensed after 1995, do the authors imply that they are better clinicians?

Response: We apologize for the verbatim. We didn’t intend to imply that. This sentence has been removed.

---

## [Decision Letter · Decision Letter 1]

26 Apr 2022

PONE-D-21-13847R1Recognition & Management of Varicella Infections and Accuracy of Antimicrobial Recommendations: Case vignettes study in the USPLOS ONE

Dear Dr. Harley and Dr. Pawaskar,

Thank you for submitting your manuscript to PLOS ONE. We recognize the substantial edits and appreciate the resubmission of your manuscript.  The reviewers felt that all comments have been addressed in the revision, but offered very minor suggestions to improve clarity. Therefore, we invite you to submit a revised version of the manuscript that addresses the points raised during the review process. Please submit your revised manuscript by June 10, 2022. If you will need more time than this to complete your revisions, please reply to this message or contact the journal office at plosone@plos.org. Please include the following items when submitting your revised manuscript:A rebuttal letter that responds to each point raised by the academic editor and reviewer(s). You should upload this letter as a separate file labeled 'Response to Reviewers'.A marked-up copy of your manuscript that highlights changes made to the original version. You should upload this as a separate file labeled 'Revised Manuscript with Track Changes'.An unmarked version of your revised paper without tracked changes. You should upload this as a separate file labeled 'Manuscript'.If applicable, we recommend that you deposit your laboratory protocols in protocols.io to enhance the reproducibility of your results. Protocols.io assigns your protocol its own identifier (DOI) so that it can be cited independently in the future. For instructions see: https://journals.plos.org/plosone/s/submission-guidelines#loc-laboratory-protocols. Additionally, PLOS ONE offers an option for publishing peer-reviewed Lab Protocol articles, which describe protocols hosted on protocols.io. Read more information on sharing protocols at https://plos.org/protocols?utm_medium=editorial-email&utm_source=authorletters&utm_campaign=protocols.

We look forward to receiving your revised manuscript.

Kind regards,

Anne Lachiewicz

Guest Editor

PLOS ONE

Journal Requirements:

Reviewers' comments:

Reviewer's Responses to Questions

**Comments to the Author**

1. If the authors have adequately addressed your comments raised in a previous round of review and you feel that this manuscript is now acceptable for publication, you may indicate that here to bypass the “Comments to the Author” section, enter your conflict of interest statement in the “Confidential to Editor” section, and submit your "Accept" recommendation.

Reviewer #1: All comments have been addressed

Reviewer #2: All comments have been addressed

2. Is the manuscript technically sound, and do the data support the conclusions?

Reviewer #1: (No Response)

Reviewer #2: Yes

3. Has the statistical analysis been performed appropriately and rigorously? 

Reviewer #1: (No Response)

Reviewer #2: N/A

4. Have the authors made all data underlying the findings in their manuscript fully available?

Reviewer #1: (No Response)

Reviewer #2: Yes

5. Is the manuscript presented in an intelligible fashion and written in standard English?

Reviewer #1: (No Response)

Reviewer #2: Yes

6. Review Comments to the Author

Reviewer #1: (No Response)

Reviewer #2: The authors have addressed all the reviewers concerns and provided major revisions to the manuscript that make it much more readable.

I have included the minor revisions below:

1. Table 4 Vignette 7 – the word pneumonia is missing

2. Table 5 – should include the baseline comparison of what is the guideline recommended antiviral/antibiotic use

3. Second to last sentence of results – remove word recommendations

4. Last sentence of results – rewrite sentence to avoid “less incorrectly” which is confusing

5. Third sentence of discussion needs to be rewritten for clarity

7. PLOS authors have the option to publish the peer review history of their article (what does this mean?). If published, this will include your full peer review and any attached files.

Reviewer #1: No

Reviewer #2: No

---

## [Author Response · Author response to Decision Letter 1]

23 May 2022

Review Comments to the Author & Responses

Thank you to the reviewers for taking time to review the paper thoroughly and providing such insightful comments. Please see our responses below. We hope that we have addressed them to your satisfaction.

Best regards,

Authors

Reviewer #2:

1. Table 4 Vignette 7 – the word pneumonia is missing

Response: The word “pneumonia” has been added to the box in column one.

2. Table 5 – should include the baseline comparison of what is the guideline recommended antiviral/antibiotic use

Response: The table has been replaced with a new table that includes baseline comparisons for full context.

3. Second to last sentence of results – remove word recommendations

Response: The word “recommendations” has been removed from the relevant sentence.

4. Last sentence of results – rewrite sentence to avoid “less incorrectly” which is confusing

Response: The sentence has been reworded and now reads as follows: “Conversely, antivirals were incorrectly recommended less often by HCPs licensed earlier (63.9%) compared to those licensed later (72.1%).”

5. Third sentence of discussion needs to be rewritten for clarity

Response: The sentence has been reworded and now reads as follows: “However, vaccine hesitancy (15) and delays in routine pediatric vaccination due to COVID-19 (11) has impacted the vaccination patterns and coverage rates. It is important to accurately diagnose and manage pediatric infectious diseases such as varicella.”

---

## [Editor Report · Decision Letter 2]

25 May 2022

Recognition & Management of Varicella Infections and Accuracy of Antimicrobial Recommendations: Case vignettes study in the US

PONE-D-21-13847R2

Dear Dr. Harley,

We’re pleased to inform you that your manuscript has been judged scientifically suitable for publication and will be formally accepted for publication once it meets all outstanding technical requirements.

Kind regards,

Anne Lachiewicz

Guest Editor

PLOS ONE

Additional Editor Comments (optional):

Thank for addressing all reviewer comments.
---

## [Editor Report · Acceptance letter]

1 Jun 2022

PONE-D-21-13847R2 

Recognition & Management of Varicella Infections and Accuracy of Antimicrobial Recommendations: Case vignettes study in the US 

Dear Dr. Harley:

I'm pleased to inform you that your manuscript has been deemed suitable for publication in PLOS ONE. Congratulations! Your manuscript is now with our production department. 

Kind regards, 

on behalf of

Dr. Anne Lachiewicz 

Guest Editor

PLOS ONE